# Itaconate stabilizes CPT1a to enhance lipid utilization during inflammation

**Rabina Mainali[1], Nancy Buechler[1], Cristian Otero[1], Laken Edwards[1], Chia-Chi Key[2], Cristina Furdui[2], Matthew A Quinn[1,2]\***

[1]Department of Pathology, Section on Comparative Medicine, Wake Forest School of Medicine, Winston Salem, United States; [2]Department of Internal Medicine, Section on Molecular Medicine, Wake Forest School of Medicine, Winston Salem, United States

**Abstract** One primary metabolic manifestation of inflammation is the diversion of cis-aconitate within the tricarboxylic acid (TCA) cycle to synthesize the immunometabolite itaconate. Itaconate is well established to possess immunomodulatory and metabolic effects within myeloid cells and lymphocytes, however, its effects in other organ systems during sepsis remain less clear. Utilizing *Acod1* knockout mice that are deficient in synthesizing itaconate, we aimed to understand the metabolic role of itaconate in the liver and systemically during sepsis. We find itaconate aids in lipid metabolism during sepsis. Specifically, *Acod1* KO mice develop a heightened level of hepatic steatosis when induced with polymicrobial sepsis. Proteomics analysis reveals enhanced expression of enzymes involved in fatty acid oxidation in following 4-octyl itaconate (4-OI) treatment in vitro. Downstream analysis reveals itaconate stabilizes the expression of the mitochondrial fatty acid uptake enzyme CPT1a, mediated by its hypoubiquitination. Chemoproteomic analysis revealed itaconate interacts with proteins involved in protein ubiquitination as a potential mechanism underlying its stabilizing effect on CPT1a. From a systemic perspective, we find itaconate deficiency triggers a hypothermic response following endotoxin stimulation, potentially mediated by brown adipose tissue (BAT) dysfunction. Finally, by use of metabolic cage studies, we demonstrate *Acod1* KO mice rely more heavily on carbohydrates versus fatty acid sources for systemic fuel utilization in response to endotoxin treatment. Our data reveal a novel metabolic role of itaconate in modulating fatty acid oxidation during polymicrobial sepsis.

**\*For correspondence:**
matt.a.quinn85@gmail.com

**Competing interest:** The authors declare that no competing interests exist.

## eLife assessment

This work describes a connection between inflammation and metabolism, in which itaconate stabilizes the mitochondrial fatty acid uptake enzyme Cpt1a to enhance fatty acid oxidation. The mechanism for itaconate action may be generalizable to other protein targets. This is an **important** advance, which is supported by **solid** experimental data.

## Introduction

Sepsis is described as a life-threatening organ dysfunction caused by a dysregulated host response to infections (*Singer et al., 2016*). Our understanding of sepsis has shifted to incorporate metabolic dysfunction as a central component of its pathogenesis. Inflammation-driven metabolic reprogramming and its consequences in the immune compartment have been investigated extensively (*Hu et al., 2022*; *Arner and Rathmell, 2023*; *Mohammadnezhad et al., 2022*). However, our understanding of metabolic derangements in central organs like the liver is limited. We have previously shown the liver is a target for profound transcriptional and metabolic remodeling in response to polymicrobial

sepsis. Specifically, we observe an alteration in mitochondrial function, TCA cycle remodeling, and hepatic lipid accumulation following sepsis (*Mainali et al., 2021*). Additionally, we have extended these findings to show that hepatic metabolic dysregulation contributes to altered systemic metabolism during sepsis (*Oh et al., 2022*). Our previous work is consistent with human studies indicating hepatic steatosis is induced during sepsis and an independent predictor of 30 day mortality (*Hou et al., 2021*; *Koskinas et al., 2008*; *Guirgis et al., 2021*). Furthermore, inhibition of the master lipid sensor peroxisome proliferator-activated receptor alpha (PPARα) within hepatocytes exacerbates sepsis-induced pathology (*Paumelle et al., 2019*). Collectively, these studies highlight the essential role of hepatic lipid metabolism in maintaining organismal adaptations to sepsis. However, molecular regulators coordinating hepatic metabolic responses to sepsis remain largely unknown.

Immune response and metabolic alterations are coupled during inflammation. Of particular interest, is the extensive reprogramming of mitochondrial metabolism in cells of myeloid lineage favoring the production of metabolites with immunomodulatory properties (*Zuo and Wan, 2019*). Itaconate is one such metabolite produced by the decarboxylation of cis-aconitate via aconitase decarboxylase 1 (Acod1), also known as immuno-responsive gene 1 (Irg1) (*Bentley and Thiessen, 1957*; *Bonnarme et al., 1995*). Numerous studies have shown itaconate exerts anti-inflammatory and anti-oxidative effects via multiple mechanisms including the induction of Nrf2[14] and ATF3 (*Bambouskova et al., 2018*), as well as inhibition of succinate dehydrogenase (SDH) (*Lampropoulou et al., 2016*), NLRP3 inflammasome (*Hooftman et al., 2020*), glycolysis (*Peace and O'Neill, 2022*; *Liao et al., 2019*). Additionally, the therapeutic potential of itaconate derivatives has shown promise in a variety of pre-clinical models of inflammatory diseases (*Peace and O'Neill, 2022*).

While studies have focused on the role of itaconate's within the immune compartment, its role in metabolically active tissues such as the liver is less defined. We have previously demonstrated sepsis elicits significant accumulation of itaconate within hepatocytes (*Mainali et al., 2021*). Utilizing *Acod1* knockout mice we aimed to address the effects of itaconate on hepatic and systemic metabolism in response to polymicrobial sepsis.

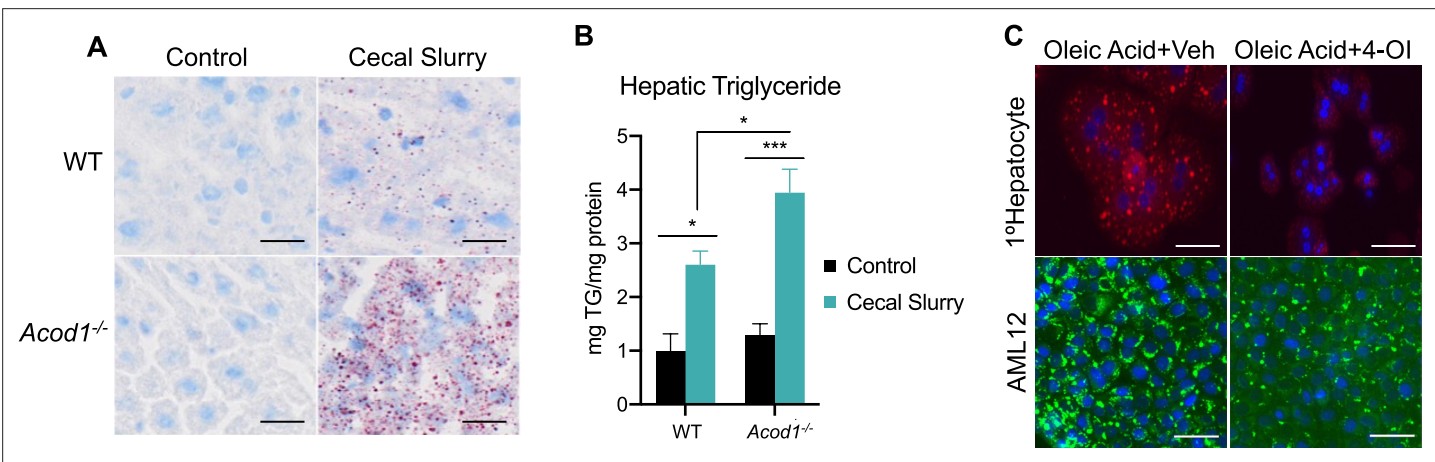

**Figure 1.** *Aconitase decarboxylase 1 (Acod1)* deficiency exacerbates hepatic lipid accumulation during sepsis. (**A**) Oil red O staining of liver sections of wild-type (WT) and *Acod1* KO control and cecal slurry (CS) injected mice (5 µl/kg). Images are representative of five independent experiments. (**B**) Hepatic triglyceride content. n=7 mice per group. (**C**) Oleate-loaded primary hepatocytes (top panel) and AML12 cells (bottom panel) were treated with vehicle (DMSO) or 4-octyl itaconate (4-OI) (250 µM) and stained with Nile Red (top panel) or BODIPY (bottom panel). Images arerepresentative of four independent experiments. Data are represented as mean ± SEM. *p<0.05, ***p<0.001. Scale bars are 50 µm.

The online version of this article includes the following figure supplement(s) for figure 1:

**Figure supplement 1.** BODIPY staining of frozen liver section of wild-type (WT) and *Acod1-/-* KO livers stained 24 hr post-LPS injection. Scale bars are 50 µm.

## Results

### *Acod1* deficiency exacerbates hepatic lipid accumulation during sepsis

We have previously reported sepsis induces a state of hepatic steatosis (*Mainali et al., 2021*). Given itaconate accumulates within hepatocytes during sepsis and previous reports demonstrating the ability of itaconate to modulate lipid metabolism (*Frieler et al., 2022*; *Li et al., 2020*; *Hall et al., 2013*; *Chu et al., 2022*), we sought to determine its role, if any, in altering the course of steatosis during sepsis. To achieve this, we subjected 8–10 weeks old male C57BL/6NJ (WT) and C57BL/6NJ-*Acod1*em1(IMPC)-J/J (*Acod1* KO) to sepsis via cecal slurry injection for 24 hr as previously described (*Gong and Wen, 2019*). To further investigate the role of itaconate in modulating hepatic lipid homeostasis during sepsis, we first evaluated the level of triglyceride accumulation. Consistent with our previous reports (*Mainali et al., 2021*), we find sepsis-induced hepatic lipid accumulation as shown by increased oil red O staining and quantification in WT mice (*Figure 1a–b*). Remarkably, septic *Acod1* KO mice exhibited enhanced lipid droplet accumulation and significantly higher triglyceride levels compared to WT litter-mates (*Figure 1a–b*). Given the aberrant steatosis observed in response to *Acod1* deficiency in male mice, we next evaluated if inflammation drives the development of hepatic steatosis in a sex-dependent manner. Parallel to male mice, female *Acod1* KO mice injected with endotoxin exhibited a higher degree of lipid droplet accumulation compared to endotoxin-treated WT littermates as demonstrated by enhanced BODIY staining in liver sections (*Figure 1—figure supplement 1*). Next, we determined if itaconate is directly modulating hepatocyte lipid metabolism or our observed phenotype is driven

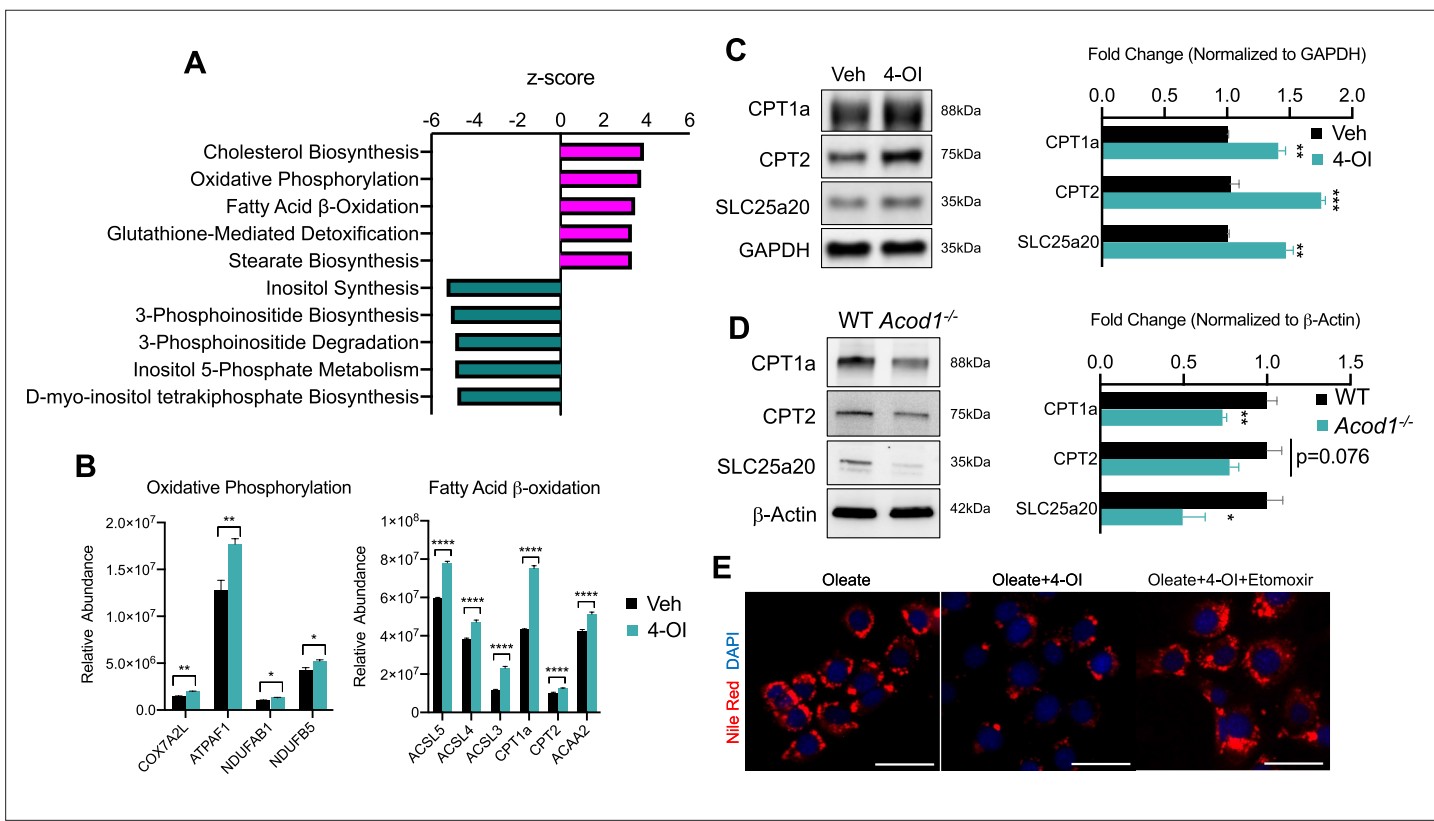

**Figure 2.** 4-octyl itaconate (4-OI) promotes mitochondrial fatty acid uptake and clearance. (**A**) Pathway analysis of significantly altered proteins from global proteomics of AML12 cells stimulated with 4-OI for 24 hr. n=5 biological replicates per group. (**B**) Quantification of CPT1a/CPT2 in proteomics analysis. (**C**) Western blot of CPT1a, CPT2, and SLC25a20 in AML12 stimulated with 4-OI for 24 hr. Quantification on the right. n=4 independent experiments. (**D**) Western blot of liver lysates of female LPS injected wild-type (WT) and *aconitase decarboxylase 1* (*Acod1)* KO mice. Quantification on the right. n=5 mice per group. (**E**) Nile Red images of lipid-loaded hepatocytes treated with 4-OI (250 μM) ± etomoxir (4 μM). n=3 independent experiments. *p<0.05, **p<0.01, ***p<0.001. Scale bars are 50 μm.

The online version of this article includes the following source data for figure 2:

**Source data 1.** Source file for Western blot 2c.

**Source data 2.** Source file for Western blot 2d.

by secondary effects such as hyperinflammation as previously reported (*Lampropoulou et al., 2016*). To achieve this, we employed an in vitro model of steatosis via oleate loading in AML12 cells and primary hepatocytes. Utilizing the cell-permeable itaconate derivative, 4-octyl itaconate (4-OI), we find that both primary hepatocytes and AML12 cells pretreated with 4-OI (250 μM) demonstrate lower oleate-induced lipid droplet formation (*Figure 1c*). Taken together, our data demonstrate itaconate acts as an anti-steatotic metabolite within the liver both in vivo and in vitro.

## 4-OI promotes mitochondrial fatty acid uptake and clearance

Our data has revealed anti-steatotic properties of itaconate, however, mechanisms conferring these actions are not resolved. To gain mechanistic insight into the anti-steatotic effects of itaconate, we performed discovery-based untargeted proteomic analysis of AML12 cells treated with 4-OI for 24 hr. Analysis of our proteomic data indicated significant regulation of several pathways related to lipid metabolism by 4-OI (*Figure 2a*). Notably, we find enhanced expression of proteins involved with oxidative phosphorylation such as COX7A2L, ATPAF1, NDUFAB1, as well as the fatty acid β-oxidation enzymes ACSL3-5, CPT1a, CPT2, and ACAA2 following 4-OI stimulation (*Figure 2b*). Enhanced expression of CPT1a, CPT2, and SLC25a20 were verified in independent experiments via western blot analysis (*Figure 2c*). Given the stimulatory effect of 4-OI on carnitine shuttle enzyme expression, we hypothesized loss of endogenous itaconate during inflammatory settings would result in impaired expression. Assessing the protein expression of these enzymes in endotoxin-treated WT and *Acod1* KO mice revealed loss of *Acod1* significantly decreased the expression of CPT1a, CPT2, and SLC25a20 (*Figure 2d*). These data reveal a stimulatory effect of 4-OI and endogenous itaconate on the expression of carnitine transport enzymes within hepatocytes. Lastly, we investigated whether the regulation of CPT1a by itaconate is linked to its anti-steatotic effects. We repeated the lipid loading experiment, however, this time we inhibited CPT1a via pharmacological inhibition with etomoxir. We find etomoxir treatment reverses the anti-steatotic effects of 4-OI resulting in lipid droplet formation similar to oleate-loaded control cells (*Figure 2e*). Collectively, our data demonstrates upregulation of β-oxidation enzymes in response to itaconate, which affords, at least in part, its anti-steatotic effects.

## 4-OI stabilizes CPT1a protein expression

The upregulation of carnitine shuttle enzyme expression in response to itaconate stimulation, to our knowledge, has not been shown before. Therefore, we aimed to determine the mechanism underlying this upregulation. Initially, we assessed transcript levels of CPT1a, CPT2, and SLC25a20 in AML12 cells stimulated with 4-OI. We find very modest induction of these genes, however, not to the extent we observe at the protein level. Furthermore, gene expression of these enzymes in liver lysates of endotoxin-stimulated WT and *Acod1* KO mice do not support the repression we observe in vivo (Data not shown). These data indicate itaconate may upregulate the expression of these enzymes at the post-translational level. Increased protein expression can be conferred through enhancing protein stability. Therefore, we first tested the stability of CPT1a in response to 4-OI stimulation in the presence of the protein synthesis inhibitor cycloheximide (CHX). CPT1a displayed a prolonged half-life of about 24 hr in vehicle-treated cells (*Figure 3a*). In contrast, stimulation with 4-OI significantly extended the half-life of CPT1a protein (*Figure 3a*). The stability of proteins is regulated at the post-translational level through activation of the ubiquitin system. Activation of E1-E3 ubiquitin ligases leads to ubiquitination of target substrates and subsequent proteasomal degradation. Given the extended half-life of CPT1a in the presence of 4-OI, we sought to determine if this is mediated via alterations in its ubiquitination status. To achieve this, we immunoprecipitated CPT1a in vehicle and 4-OI-stimulated cells in the presence of proteasome inhibitor MG132. Immunoprecipitation of CPT1a and subsequent western blotting of ubiquitin revealed little to no ubiquitination in both vehicle and 4-OI treated groups in the absence of MG132 (*Figure 3b*). However, in the presence of MG132 we observe robust polyubiquitination of CPT1a in the vehicle-treated group (*Figure 3b*). In contrast, 4-OI stimulation drastically reduced levels of CPT1a polyubiquitination (*Figure 3b*). Taken together, our data indicate 4-OI interferes with CPT1a ubiquitination promoting its stabilization.

Alkylation of cysteine residues is a post-translational modification that regulates a vast array of cellular processes. Given the highly nucleophilic nature of thiol sidechains, they can participate in Michael's reaction and undergo conjugation with molecules like itaconate that have an electrophilic α,β−unsaturated carboxylic acid to form adducts. Given that itaconate and its derivatives have been

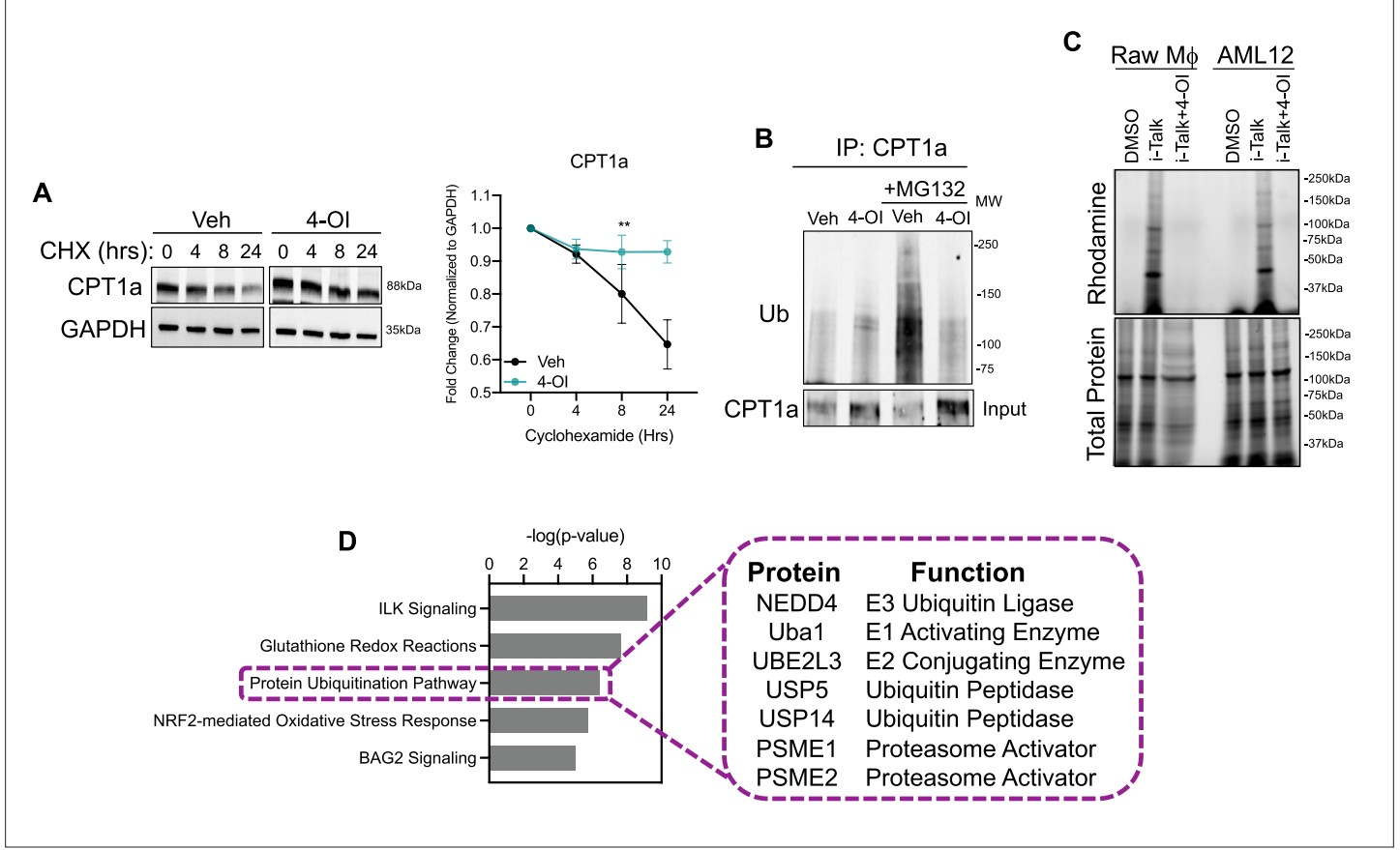

**Figure 3.** 4-octyl itaconate (4-OI) stabilizes CPT1a protein expression. (**A**) Western blot and quantification of CPT1a in AML12 cells that were pretreated with vehicle or 4-OI for 24 hr, then stimulated with cyclohexamide (CHX). Quantification on the right. n=3 independent experiments. (**B**) Immunoprecipitation (IP) of CPT1a in AML12 cells pretreated with 4-OI for 24 hr followed by stimulation with MG132 for 6 hr. Equal amounts of proteins were IP'd with anti-CPT1a and subjected to Western blot analysis with ubiquitin antibody. 5% input below. n=3 independent experiments. (**C**) In-gel fluorescence of rhodamine in iTalk labeled Raw macrophages and AML12 hepatocytes. (**D**) Pathway analysis of global proteomics of iTalk-enriched proteins in AML12 cells stimulated with ITalk for 4 hr. **p<0.01.

The online version of this article includes the following source data for figure 3:

**Source data 1.** Source file for Western blot 3a.

**Source data 2.** Source file for Western blot 3b.

shown to modulate biological function via the alkylation of numerous proteins, we next tested the hypothesis that itaconate interacts with CPT1a and hinders ubiquitination. This is based on previous reports in macrophages in which CPT1a was shown to interact with itaconate via the biorthogonal probe iTalk (*Qin et al., 2020*). To determine initially if itaconate interacts with the hepatic proteome we labeled AML12 cells with iTalk for 4 hr followed by a click reaction to an azide-rhodamine probe. In-gel fluorescence of both Raw Macrophages and AML12 cells treated with iTalk display a banding pattern compared to vehicle-treated cells (*Figure 3c*). Additionally, pre-treatment with the competitive inhibitor 4-OI blocks this banding pattern, indicating the specificity of the iTalk probe for itaconation (*Figure 3c*). These data indicate that hepatic proteins are indeed subject to itaconation. Next, we performed iTalk labeling in AML12 cells followed by subsequent azide-agarose click reaction to allow for mass spectrometry identification of hepatic itaconated proteins. We identified 123 proteins that had ≥1.5 fold enrichment over vehicle-treated cells. We initially surveyed the hepatic itaconated proteins to determine if CPT1a was enriched. Contrary to our hypothesis and previous reports, we did not identify CPT1a as a hepatic itaconation substrate. Therefore, we next performed pathway analysis to gain insight into biological pathways that may afford itaconate's ability to stabilize CPT1a protein. Consistent with previous reports, we find enrichment in proteins involved in glutathionylation and NRF2-mediated oxidative signaling (*Figure 3d*). Additionally, we found proteins involved in protein

ubiquitination to show enrichment in the itaconation group (*Figure 3d*). These proteins range from E1-E3 ubiquitin ligases as well as various components of the proteasome and ubiquitin peptidases (*Figure 3d*). Identification of these components involved in proteasomal turnover of proteins within the liver is the first to our knowledge to be demonstrated. We hypothesize itaconation of ubiquitin ligases and proteasome components may confer the stabilizing effects of itaconate on CPT1a.

### *Acod1* deficiency impairs the thermogenic program during sepsis

Apart from the liver, adipose tissues also play a central role in the maintenance of whole-body energy homeostasis. While white adipose tissues function to store excess energy in the form of triglycerides, brown adipose tissue (BAT) are metabolically active adipose depots which contribute to non-shivering thermogenesis (*Kajimura et al., 2015*). This is achieved, in part, through oxidation of fatty acids and activation of UCP1 (*Lee et al., 2015*). A decrease in body temperature during sepsis is an independent predictor of mortality (*Rumbus et al., 2017*; *Fatteh et al., 2021*). Therefore, given the

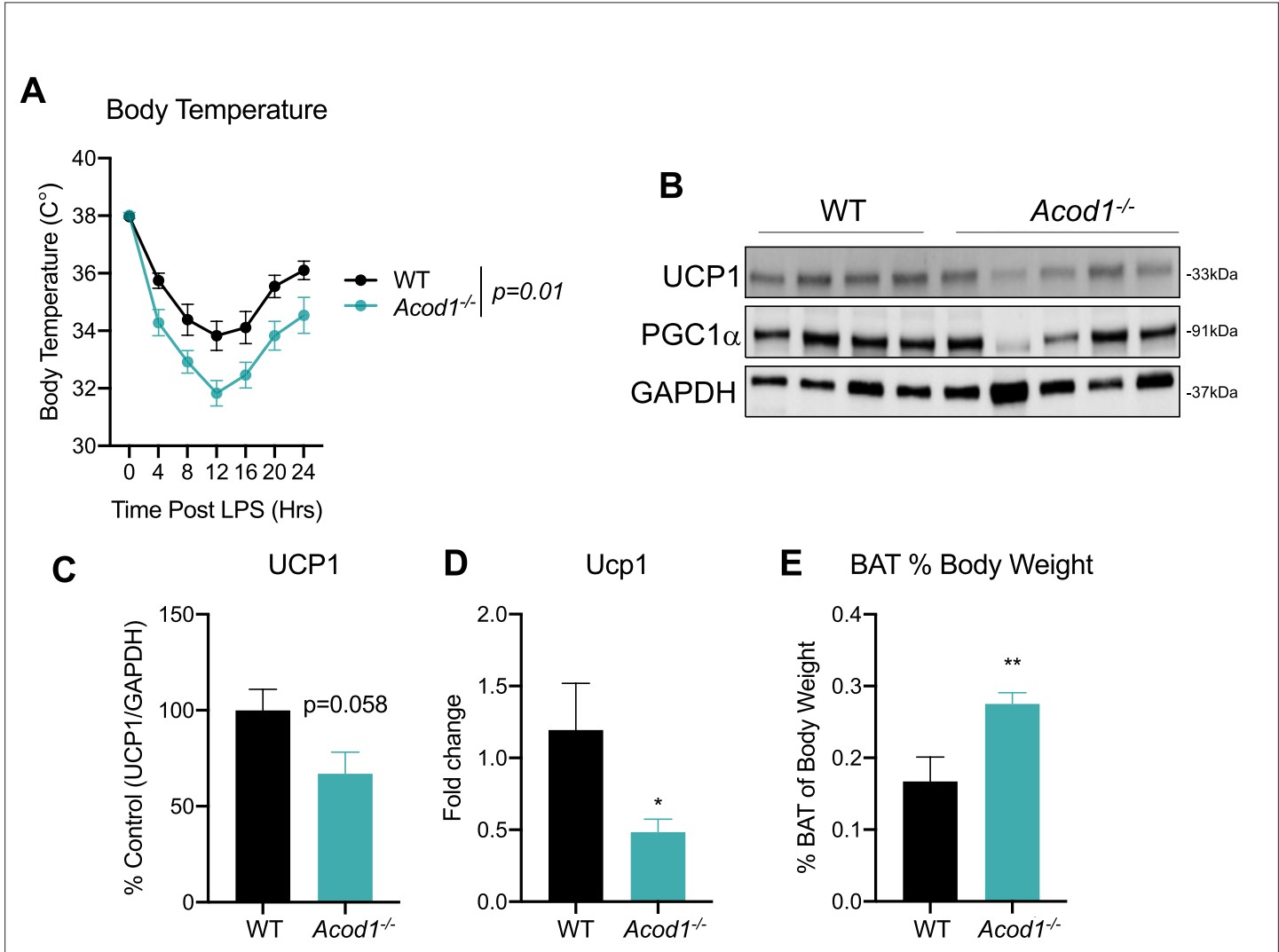

**Figure 4.** *Aconitase decarboxylase 1 (Acod1)* deficiency promotes hypothermia and brown adipose tissue (BAT) dysfunction during endotoxin challenge. (**A**) Core body temperature in female wild-type (WT) and *Acod1* KO mice following LPS injection (5 mg/kg). n=5–8 mice per group. (**B**) Western blot of UCP1, PGC-1α, and GAPDH in BAT of LPS injected WT and *Acod1* KO mice. (**C**) Quantification of UCP1 protein normalized to GAPDH. n=5–8 mice per group. (**D**) qPCR of UCP1 in BAT of LPS injected WT and *Acod1* KO mice. n=5–8 mice per group. (**E**) BAT weight 24 hr post-LPS injection in WT and *Acod1* KO mice. n=5–8 mice per group. *p<0.05, **p<0.01.

The online version of this article includes the following source data for figure 4:

**Source data 1.** Source file for Western blot 4b.

importance of thermogenesis during sepsis and our findings that *Acod1* deficiency impairs hepatic β-oxidation, we endeavored to determine if this has effects on BAT function. We assessed core body temperature in response to an endotoxin challenge in WT and *Acod1* KO mice. WT display a significant drop in body temperature in response to LPS treatment, peaking at 12 hr and returning to near baseline by 24 hr (*Figure 4a*). We find *Acod1* KO mice have a significantly more dramatic drop in body temperature following LPS treatment compared to their WT littermates (*Figure 4a*). Next, we aimed to characterize potential mechanism underlying the hypothermic phenotype in *Acod1* KO mice. To this end, we assessed the protein expression of UCP1. We find impairment in UCP1 gene and protein levels in *Acod1* deficient mice following endotoxin treatment, independent of PGC1α expression (*Figure 4b–d*). Furthermore, the BAT of endotoxin-challenged *Acod1* KO mice was larger in weight compared to LPS-injected WT septic mice (*Figure 4e*). In summary, these data indicate an impairment in UCP1-driven thermogenesis in the BAT of *Acod1*-deficient mice following an inflammatory challenge. Furthermore, our data indicate this could be mediated by impaired BAT fatty acid oxidation, as evidenced by the larger BAT depot in inflamed *Acod1* KO mice.

### *Acod1* deficiency impairs systemic substrate utilization during sepsis

We have previously reported a global shift in systemic fuel preference from glucose to fatty acid oxidation in response to CLP-induced sepsis (*Oh et al., 2022*). Our data thus far indicate itaconate deficiency impairs lipid metabolism during inflammation at the organ level. However, it remains unknown the systemic effects of *Acod1* KO on sepsis-induced shifts in fuel preference. However, it is known that *Acod1* KO mice favor glucose oxidation over fatty acids under baseline conditions (*Frieler et al., 2022*). Given this, we next investigated the effects of itaconate deficiency on inflammation-induced

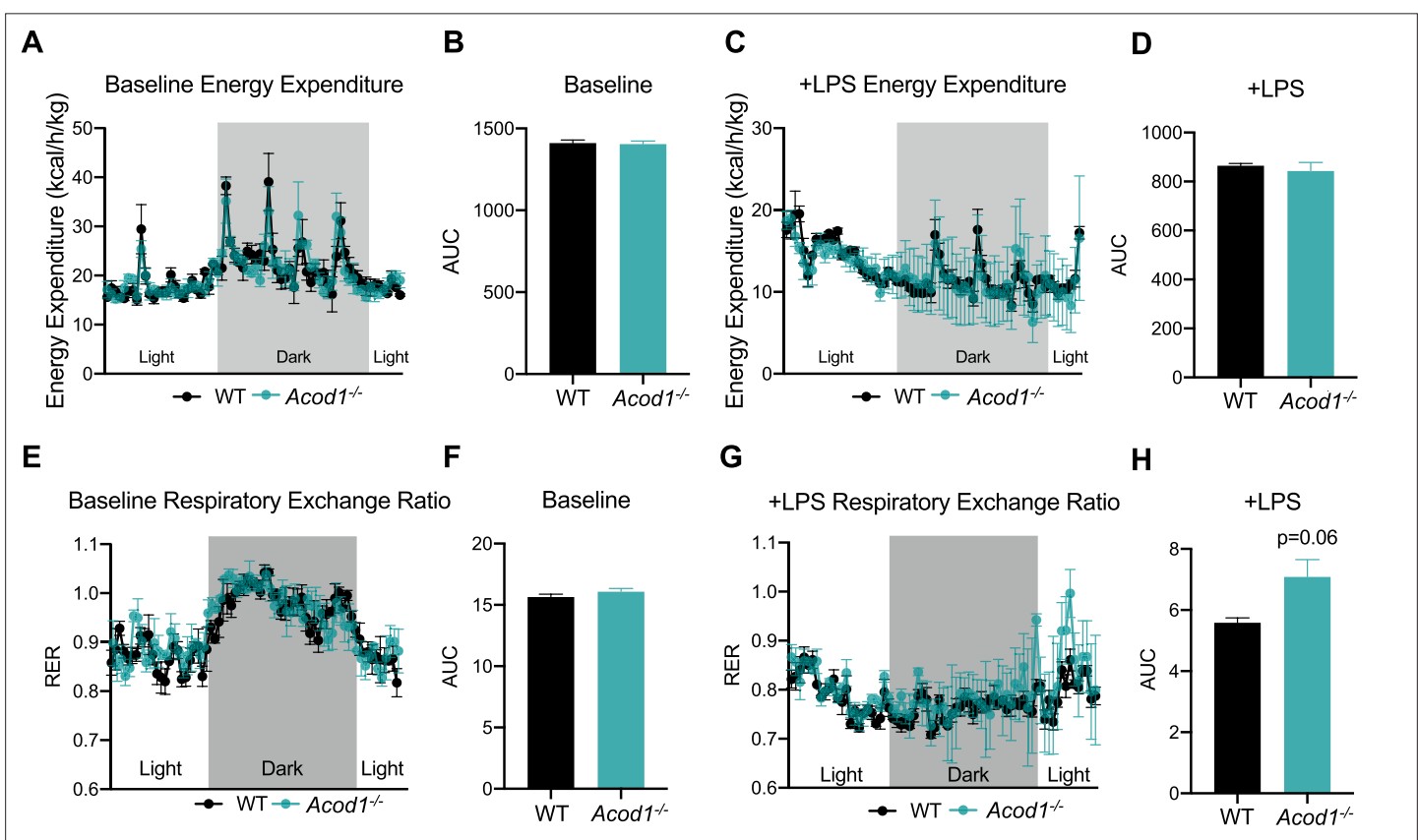

**Figure 5.** *Aconitase decarboxylase 1 (Acod1) deficiency impairs systemic substrate utilization during sepsis.* (**A**) Energy Expenditure (kcal/hr/kg) during baseline and (**C**) post-LPS injection in female *Acod1* KO and wild-type (WT) controls. Plots represent 24 hr cycle. n=3 mice per group. (**B, D**) Area under the curve for energy expenditure values over 24 hr cycle from panel B and D. (**E**) Respiratory exchange ratio (RER) during baseline and (**G**) post-LPS injection in female *Acod1* KO and WT littermate controls. Plots represent 24 hr cycle. n=3 mice per group. (**F, H**) Area under the curve for RER values over 24 hr cycle from panel F and H. Statistical significance was calculated using an unpaired two-tailed Student's t-test.

metabolic flexibility. We first assessed energy expenditure in LPS-injected WT and *Acod1* KO mice by using indirect calorimetry-enabled metabolic cages. We observed a decrease in energy expenditure in both groups in response to LPS, however, no significant differences between the groups was appreciated, under both baseline and post-LPS injection (*Figure 5a–d*). Next, we interrogated the respiratory exchange ratio (RER) as an indicator of systemic fuel preference. Consistent with our previous study (*Oh et al., 2022*), WT septic mice demonstrated a shift in fuel preference indicated by a decrease in RER from about 1–0.7 (*Figure 5e–h*), signifying a higher reliance on fatty acid oxidation over glucose utilization. Interestingly, we found *Acod1* KO mice had a decrease in RER following LPS treatment, however, not to the same extent as their WT-type counterparts (*Figure 5g, h*), indicating itaconate deficiency leads to a subtle defect of systemic fat utilization.

## Discussion

Our studies identify itaconate as a central modulator of lipid metabolism during the course of overt inflammation. Specifically, we show itaconate may enhance lipid clearing in the liver through the stabilization of mitochondrial fatty acid uptake enzyme CPT1a. Additionally, we have uncovered novel itaconate substrates involved in protein ubiquitination which may underlie the lipid-clearing effects of itaconate. Lastly, we demonstrate systemic defects in lipid metabolism and thermogenesis in itaconate-deficient mice following the endotoxin challenge. These studies extend our understanding of itaconate as a metabolic regulator in response to inflammation.

Mitochondrial bioenergetics and metabolic reprogramming play a crucial role in promoting both immune and non-immune changes in response to inflammation (*Mainali et al., 2021*; *Weinberg et al., 2015*; *Choi et al., 2021*; *Paumelle et al., 2019*; *Chouchani and Kajimura, 2019*). Furthermore, metabolic dysregulation in the context of sepsis can profoundly contribute to impaired lipid metabolism, which can enhance sepsis severity and mortality (*Oh et al., 2022*; *Amunugama et al., 2021*; *Sharma et al., 2019*; *Barker et al., 2021*; *Mainali et al., 2021*). We previously demonstrated sepsis elicits a systemic increase in fatty acid mobilization and utilization (*Mainali et al., 2021*), which supports a systemic shift in fuel preference from glucose to fatty acid oxidation (*Oh et al., 2022*). Additionally, we have shown sepsis induces dyslipidemia, which in turn promotes the development of hepatic steatosis (*Oh et al., 2022*; *Mainali et al., 2021*). While the exact mechanisms driving alterations of hepatic lipid metabolism during inflammation are not fully understood, dysregulation of hepatic PPARα signaling has been implicated (*Paumelle et al., 2019*; *Lewis et al., 2016*; *Van Wyngene et al., 2020*). These pre-clinical findings are consistent with human septic patients, which show the presence of hepatic steatosis (*Koskinas et al., 2008*; *Garofalo et al., 2019*). Our data identifies itaconate as a novel pathway for the regulation of hepatic lipid metabolism during sepsis.

The role of itaconate in modulating fatty acid oxidation is becoming more appreciated. Several previous studies have demonstrated itaconate promotes β-oxidation (*Hall et al., 2013*, *Weiss et al., 2018*). Our studies demonstrate itaconate protects against aberrant hepatic steatosis during sepsis. Furthermore, we identify the β-oxidation pathway as a target of itaconate. Interestingly, *Acod1* KO mice have been shown to have decreased fatty acid oxidation and enhanced glucose oxidation compared to wild-type mice (*Frieler et al., 2022*). Our data sheds insight into the mechanism by which this may be afforded. Specifically, the modulation of CPT1a and other carnitine shuttle enzymes underlies, at least in part, the regulatory role of itaconate on the β-oxidation pathway.

Previously, it has been reported that itaconate plays a role in regulating the β-oxidation of fatty acids to fuel OXPHOS. This has been observed in various cell types, including hepatocytes from mouse models of NAFDL, tissue-resident macrophages in peritoneal tumors (*Weiss et al., 2018*), macrophages from zebrafish (*Hall et al., 2013*) as well as T cell subsets Th17- and Treg- polarizing T cells (*Aso et al., 2023*). This is in conjunction with our observations given significant upregulation in proteins governing the β-oxidation and OXPHOS pathway by itaconate. Interestingly, we also observed itaconate regulation of key steps within cholesterol biosynthesis in ALM12 cells when assessing pathways involved in cellular metabolism. We hypothesize this is due to the inactivation of vitamin $B_{12}$ by itaconyl-CoA, resulting from itaconate activation (*Cordes and Metallo, 2021*; *Shen et al., 2017*). This can cause impairment of methionine synthase activity, an enzyme dependent on vitamin $B_{12}$, leading to dysregulation in the conversion of homocysteine to methionine, and ultimately, alterations in the abundance of S-adenosylmethionine (SAM), a methyl donor in numerous biological and biochemical processes (*Froese et al., 2019*). Interestingly deficiency in $B_{12}$ has been found to

induce cholesterol biosynthesis by limiting SAM and modulating the methylation of SREBF1 and LDLR genes (*Adaikalakoteswari et al., 2015*). Additionally, alteration in intracellular SAM abundance and reduction in SAM/SAH ratio by itaconate was found to influence TH17 /Treg cell differentiation as a result of tri-methylation of histone H3 protein-induced epigenetic reprogramming (*Aso et al., 2023*). However, the exact mechanism by which itaconate regulates metabolic and epigenetic reprogramming to enhance hepatic cholesterol synthesis is not known and needs to be further investigated given the crucial role of cholesterol in maintaining cellular organization, steroid hormone, bile acids, and vitamin D synthesis (*Röhrl and Stangl, 2018*).

Frieler et al. recently uncovered a protective role of endogenous itaconate in regulating adipocyte metabolism through altering glucose homeostasis, lipolysis, and adipogenesis during HFD-induced obesity (*Frieler et al., 2022*). Furthermore, they demonstrated endotoxin induces the expression of *Acod1* in brown adipose tissue (BAT). However, the biological function of inflammation-induced *Acod1* in BAT was not studied. A previous study demonstrated improved body temperature and clinical scores in septic mice upon exogenous 4-OI administration (*Mills et al., 2018*). However, the role of endogenous itaconate in modulating body temperature in response to inflammation has yet to be investigated. Our study begins to fill this gap by showing a pro-thermogenic effect of itaconate during an endotoxin challenge. While the mechanism underlying these effects were not investigated in the current study, we show a defect in the primary thermogenic mediator UCP1. Research has demonstrated that inter-organ crosstalk between BAT and the liver is essential to elicit non-shivering thermogenesis. Specifically, BAT utilizes hepatic-derived acylcarnitines released in response to cold stress (*Simcox et al., 2017*), Intriguingly, these studies demonstrated a reliance on hepatic CPT1 function to mediate cold stress-induced acylcarnitine secretion (*Simcox et al., 2017*). Our data demonstrates impaired CPT1a expression in the liver of endotoxin-challenged *Acod1* KO mice. These findings in conjunction with previous literature may provide a potential physiological mechanism by which itaconate deficiency hinders hepatic acylcarnitine production due to impairment in CPT1a expression. Future studies targeting hepatic CPT1a expression in *Acod1* KO mice would uncover causal links between hepatic fatty acid oxidation and thermogenesis in response to endotoxin. This is important, as clinically and in murine models of sepsis, the onset of hypothermia is observed and considered a predictor of mortality (*Lewis et al., 2016*; *Kushimoto et al., 2013*).

Given its electrophilic properties and its ability to directly alkylate cysteine residues, we initially hypothesized regulation of FAO was driven by enhancement of CPT1a due to its itaconation. Contrary to our hypothesis, proteomic profiling of ITalk substrates in AML12 cells did not reveal CPT1a as a target of itaconation. However, proteomic analysis revealed enrichment of the protein ubiquitination pathway by ITalk. We found this of interest, as this may explain the stabilizing effects of itaconate on CPT1a protein expression. The regulation of protein ubiquitination by itaconate is a mechanism that has been implicated in numerous models of inflammation. The first is the classical upregulation of NRF2 due to the alkylation of cysteine residues of KEAP1[23], which functions as an adaptor of the Cul3-based ubiquitin E3 ligase complex. This covalent modification promotes the dissociation of KEAP1 from CUL3 to inhibit the conjugation of ubiquitin onto the N-terminal domain of NRF2 (*Mills et al., 2018*; *Yamamoto et al., 2018*; *Canning et al., 2015*). Additionally, 4-OI has been shown to negatively regulate osteoclastogenesis and inflammatory response by suppressing E3 ubiquitin-protein ligase HRD1 to activate Nrf2 signaling (*Sun et al., 2019*). However, the mechanism underlying the inhibition of HRD1 by itaconate was not investigated or discussed. Our data extends these findings and indicates an interaction between itaconate and multiple components involved in proteasomal turnover of proteins. Future studies aimed at dissecting the regulatory role of these enzymes in the proteasomal turnover of CPT1a and the anti-steatotic effect of itaconate are warranted.

Accumulation of itaconate in the mouse liver during sepsis has highlighted the necessity to understand its functional role within this compartment. By endeavoring to uncover the biological role of itaconate in hepatocytes, we have uncovered a novel function of itaconate within the liver and systemically, to aid in fatty acid processing in the face of inflammation. Furthermore, our work identifies a potentially new mechanism of action via the stabilization of CPT1a. Finally, our work has uncovered the systemic effects of endogenous itaconate on metabolic flexibility and thermogenesis in response to inflammation. Overall, these findings suggest that interventions aimed at regulating the *Acod1*/ itaconate axis may hold potential therapeutic advantages in regulating dyslipidemia at both the local

and systemic levels observed during sepsis. Future work is necessary to understand cellular sources of itaconate, and the role of this immunometabolite in coordinating interorgan crosstalk during sepsis.

## Materials and methods

### Animals experiments

Male and female *Acod1* KO and WT littermates aged 8–10 weeks were purchased from The Jackson Laboratory (*Acod1*[em1(IMPC)J]/J) (Bar Harbor, ME). All animals were subject to a 12:12 hr dark/light cycle with ad libitum access to standard rodent chow and water. To induce sepsis, cecal slurry (CS) (5 µl/kg) was injected as previously described (*Starr et al., 2014*). For endotoxin-induced sepsis female WT and *Acod1* KO mice were injected with LPS (5 mg/kg). Tissues were harvested 24 hr after either CS or LPS injection. All experiments and procedures involving mice were carried out following the approved protocols of the Institutional Animal Care and Use Committee (IACUC) of Wake Forest School of Medicine. Animals were randomly assigned to experimental or control groups.

### Histological analysis and lipid droplet staining

After undergoing treatments, the cells were rinsed with PBS and fixed in 4% PFA at room temperature for 5 min. Subsequently, the cells were stained with Nile Red for 10 min at 37 °C, washed with PBS, and mounted using Fluroshied with DAPI. The cells were visualized using the ZOETM Fluorescent Cell Imager (Bio-Rad, 1450031). Liver sections with a thickness of 5 µm, were fixed with 4% PFA for 10 min at room temperature. Next, the slides were dipped a few times in 60% isopropyl alcohol and then incubated in the working solution of Oil Red O for 10 min. The slides were subsequently rinsed a few times in 60% isopropyl alcohol, followed by three rinses with distilled water. The sections were then stained with hematoxylin as a counterstain for 1 min, followed by three rinses with distilled water. Finally, the slides were mounted using Fluroshied with DAPI and imaged. When reviewing histological slides reviewers were blinded to treatment groups.

### Triglyceride measurement

Hepatic triglyceride content was determined via a colorimetric assay kit according to the manufacturer's protocol (Abcam).

### Hepatocyte isolation

Primary mouse hepatocytes were isolated via portal vein perfusion and collagenase digestion as previously described (*Chen et al., 2000*) from male WT C57BL6J mice. After perfusion, hepatocytes cells were liberated by dissociation in DMEM (Thermo Fisher; CA, USA). Cells were then filtered through nylon mesh to remove cellular debris and connective tissue and the resulting cells were pelleted by centrifugation at 50 g for 1 min. Pellets were washed three times with DMEM and viability was assessed via Trypan Blue exclusion.

### Cell culture

Murine hepatocyte cell line AML12 (ATCC, #CRL-2254) were maintained on plastic cell culture plates in Dulbecco's Modified Eagle Medium/Nutrient Mixture F12 (DMEM/F-12) supplemented with 10% FBS, Gibco Insulin-Transferrin-Selenium supplement (Gibco), dexamethasone, and penicillin and streptomycin in a humidified incubator (37 °C, 5% $CO_2$). Reagents added to cell culture media are as follows: 250 µM 4-octyl itaconate (Tocris Bioscience, 6662), 4 µM Etomoxir (Sigma, E1905), Oleate (Sigma, O1008), Cyclohexamide (Sigma, C4859), 10 µM MG132 (Cell Signaling 2194 S).

### Antibodies

Primary antibodies used in the study are as follows: mouse monoclonal anti-GAPDH (Santa Cruz Biotechnology, SC-32233), rabbit polyclonal anti-ACSL1 (Protein Tech, 13989), mouse monoclonal anti-CPT1a (Protein Tech, 151841), rabbit polyclonal anti-CPT2 (Protein Tech, 2655), rabbit polyclonal anti-SLC25A20 (Protein Tech, 19363), mouse monoclonal anti-Ubiquitin (Cell Signaling Technology, 3936), and mouse monoclonal anti-β-actin (Cell Signaling Technology, 3700 S).

## Western blot and co-immunoprecipitation

Protein lysates were prepared from the livers of mice by homogenization in SDS sample buffer (Biorad, Hercules, CA) containing β-mercaptoethanol (Sigma) or cell scraping in AML12 cells. Approximately 30 µg of total protein was resolved on a 4–20% Tris-glycine gel (Biorad) and transferred onto a 0.2 mM nitrocellulose membrane (Biorad). Membranes were blocked with blocking buffer (LI-COR Biosciences, Lincoln, NE) and incubated overnight with primary antibodies as indicated. Secondary antibodies IRDye 800CW Goat anti-Mouse IgG (LI-COR, 926–32210) and IRDye 680RD Goat anti-Rabbit IgG (LI-COR, 926–68071) were used to detect proteins of interest via the ChemiDoc MP Imaging System (Biorad).

## Co-immunoprecipitation

Co-immunoprecipitation experiments were performed utilizing Pierce MS Compatible Magnetic IP Kit (Thermo Fisher Scientific, 90409). 1 mg of total protein was incubated with 5 ug of anti-CPT1a overnight at 4 °C, then incubated with pierce protein A/G magnetic beads for an hour at room temperature. Beads were washed and then boiled for 7 min in 1 X Laemmli SDS sample buffer. Proteins were analyzed using western blotting with an anti-ubiquitin antibody and imaged.

## RNA isolation and RT-qPCR

One hundred nanograms of total RNA was reverse-transcribed (RT) and amplified using the iScript One-Step RT-PCR kit for probes (Bio-Rad, Hercules, CA). Real-time qPCR was performed with the Bio-Rad CFX96 sequence detection system using predesigned primer/probe sets against CPT1a, CPT2, and SLC25a20 from Applied Biosystems (Foster City, CA). The relative fluorescence signal was normalized to PPIB using the ddCT method (*Livak and Schmittgen, 2001*).

## BSA-oleate complex

0.25 M of oleic acid (OA) in 100% ethanol and 0.5% BSA in DPBS was prepared and incubated in 60 °C water bath for 30 min. 800 uL of 0.25 M OA was added dropwise to 49.2 mL of 0.5% BSA to make 4 mM OA in 5% BSA. The solution was heated for an additional 3 hr with vigorous vortexing every 30 min until the solution was clear. OA BSA conjugate was warmed for 30 min in a 60 °C water bath before cell treatment.

## iTalk and click chemistry

AML12 cells were treated with 100 µM Itaconate-alkyne (iTalk, MedChemExpress, HY-133870) for 4 hr. Cells were then lysed and clicked to either rhodamine azide (Click Chemistry Tools) for in-gel fluorescence, or agarose azide for enrichment. Enriched proteins were eluted for mass spectrometry analysis according to the Click-&-Go Dde Protein Enrichment Kit (Click Chemistry Tools, 1153).

## MS/MS analysis

Samples were analyzed on an LC-MS/MS system consisting of an Orbitrap Eclipse Mass Spectrometer (Thermo Scientific, Waltham, MA) and a Vanquish Neo nano-UPLC system (Thermo Scientific, Waltham, MA). Peptides were separated on a DNV PepMap Neo (1500 bar, 75 µm × 500 mm) column for 120 min employing linear gradient elution consisting of water (A) and 80% acetonitrile (B) both of which contained 0.1% formic acid. Data were acquired by top speed data-dependent mode where maximum MS/MS scans were acquired per cycle time of 3 s between adjacent survey spectra. MS2 scans were repeated with precursor ion subsets isolated by ion mobility using the FAIMS which compensation voltage was set to –45 eV, –55 eV, and –65 eV sequentially. Dynamic exclusion option was enabled where duration was set to 120 s. To identify proteins, spectra were searched against the UniProt mouse protein FASTA database (20,309 annotated entries, Jun 2021) using the Sequest HT search engine with the Proteome Discoverer v2.5 (Thermo Scientific, Waltham, MA). Search parameters were as follows: FT-trap instrument; parent mass error tolerance, 10 ppm; fragment mass error tolerance, 0.6 Da (monoisotopic); enzyme, trypsin (full); # maximum missed cleavages, 2; variable modifications, +15.995 Da (oxidation) on methionine; static modification (only for soluble part), +57.021 Da (carbamidomethyl) on cysteine. The mass spectrometry proteomics data have been deposited to the ProteomeXchange Consortium via the PRIDE (*Perez-Riverol et al., 2022*) partner repository with the dataset identifier PXD047706 and 10.6019/PXD047706.

## Metabolic cage studies

Metabolic cages (TSE PhenoMaster system) were used in awake mice to simultaneously measure oxygen consumption, carbon dioxide production, respiratory exchange ratio, energy expenditure, food/water intake, and activity during a 12 hr light/12 hr dark cycle for five consecutive days as previously described (*Seramur et al., 2023*).

## Statistical analysis

Statistics were performed with GraphPad Prism v8. When comparing two groups an unpaired Student's two-tailed t-test was performed. When comparing three groups or more a one-way ANOVA was performed. Data are represented as mean ± SEM. Data are comprised of individual biological replicates. Group sizes were determined based on previously published work (*Mainali et al., 2021*).

## Acknowledgements

We wish to acknowledge the support of the Metabolic Phenotyping Shared Resource supported by Wake Forest CTSI. The authors would like to also acknowledge the intellectual support provided by the Center for Redox Biology and Medicine at Wake Forest School of Medicine. The authors wish to acknowledge the support of the Wake Forest Baptist Comprehensive Cancer Center Proteomics and Metabolomics Shared Resource, supported by the National Cancer Institute's Cancer Center Support Grant (P30CA012197), and of the pre-doctoral Redox Biology and Medicine Training grant (T32GM127261).

## Additional information

### Funding

| Funder | Grant reference number | Author |
|---|---|---|
| Wake Forest School of Medicine | Start-Up Funds | Matthew A Quinn |
| National Cancer Institute | P30CA012197 | Cristina Furdui |
| National Institute of General Medical Sciences | T32GM127261 | Rabina Mainali Cristina Furdui |

The funders had no role in study design, data collection and interpretation, or the decision to submit the work for publication.

### Author contributions

Rabina Mainali, Conceptualization, Data curation, Formal analysis, Investigation, Writing – original draft; Nancy Buechler, Data curation, Methodology, Project administration; Cristian Otero, Laken Edwards, Data curation; Chia-Chi Key, Cristina Furdui, Data curation, Methodology; Matthew A Quinn, Conceptualization, Data curation, Formal analysis, Supervision, Funding acquisition, Investigation, Visualization, Methodology, Writing – original draft, Project administration

### Author ORCIDs

Chia-Chi Key ⓘ http://orcid.org/0000-0003-0669-2936
Matthew A Quinn ⓘ https://orcid.org/0000-0002-3528-6569

### Ethics

All animal work performed in these studies were in accordance and approval of the institution animal care and use committee (IACUC protocol A23-094). Every effort was made to minimize suffering of animals.

Reviewer #1 (Public Review): https://doi.org/10.7554/eLife.92420.2.sa1
Reviewer #2 (Public Review): https://doi.org/10.7554/eLife.92420.2.sa2
Author Response: https://doi.org/10.7554/eLife.92420.2.sa3

## Additional files

### Supplementary files
• MDAR checklist

### Data availability
The mass spectrometry proteomics data have been deposited to the ProteomeXchange Consortium via the PRIDE partner repository with the dataset identifier PXD047706.

The following dataset was generated:

| Author(s) | Year | Dataset title | Dataset URL | Database and Identifier |
|---|---|---|---|---|
| Lee J, Matthew AQ | 2024 | Itaconate stabilizes CPT1a to enhance lipid utilization during inflammation | https://www.ebi.ac.uk/pride/archive/projects/PXD047706 | PRIDE, PXD047706 |

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
