## [Editor Report · eLife assessment]

This work describes a connection between inflammation and metabolism, in which itaconate stabilizes the mitochondrial fatty acid uptake enzyme Cpt1a to enhance fatty acid oxidation. The mechanism for itaconate action may be generalizable to other protein targets. This is an **important** advance, which is supported by **solid** experimental data.

---

## [Referee Report · Reviewer #1 (Public Review)]

Summary:

Mainali and colleagues provide evidence for Itaconate stabilising Cpt1a via a decrease in ubiquitination. This in turn likely regulates fatty acid oxidation which in turn would appear to be involved in thermoregulation in the context of sepsis.

Strengths:

These findings add to our knowledge of the role of Itaconate in sepsis and its rather complex effects on metabolism, specifically lipid metabolism.

Weaknesses:

1. This is a complex paper and would benefit from a schematic depicting the key findings.

2. The paper would benefit from additional supporting evidence. Would it be possible to measure fatty acid oxidation by metabolic tracing here, in IRG-deficient cells or in response to 4-OI? Although changes in protein level for Cpt1A are seen, this is correlated with fatty acid oxidation rather than direct demonstration. This may be challenging but would strengthen the manuscript.

3. The aspect concerning body temperature regulation is confusing. Would Itaconate not promote fatty acid oxidation to increase or maintain body temperature? Itaconate must therefore not be involved in the hypothermic response? Bringing UCP1 into the finding is confusing and needs to be better explained. Again a diagram would help, but enhanced BAT fatty acid oxidation and UCP1 expression appear linked here, with both being affected by Itaconate. This needs clarifying.

---

## [Referee Report · Reviewer #2 (Public Review)]

Summary:

This manuscript provides important new findings regarding the connection between inflammation and metabolism. It also identifies a new type of post-translational modification and its connection to protein stability. This finding is expected to be generalizable to other protein targets. In vitro evidence is solid. In vivo evidence needs some additional controls.

Strengths:

A new connection between inflammation and metabolism.

A novel type of PTM was identified.

Findings would be of broad interest and the mechanisms are likely generalizable to related control systems.

In vitro data are well-supported.

The authors successfully demonstrated that treatment with 4-octyl Itaconate (4-OI), a prodrug form of itaconate, reduces neutral lipid accumulation in the AML12 cell line and primary hepatocytes. They show that 4-OI promotes fatty acid beta-oxidation through increased stability of CPT1a protein, the rate-limiting step in this process.

Weaknesses:

Some conclusions involving the Irg1 knockout mice require important controls and clarifications to be fully convincing and some controls are missing.

---

## [Author Response]

**Reviewer #1:**
1. This is a complex paper and would benefit from a schematic depicting the key findings.

This comment is appreciated. Unfortunately, due to time restraints, the authors were not able to graphically depict our findings.

1. The paper would benefit from additional supporting evidence. Would it be possible to measure fatty acid oxidation by metabolic tracing here, in IRG-deficient cells or in response to 4-OI? Although changes in protein level for Cpt1A are seen, this is correlated with fatty acid oxidation rather than direct demonstration. This may be challenging but would strengthen the manuscript.

This is a great comment. While we did not directly measure fatty acid flux in our manuscript, Weiss et al. Nature Metabolism 2023 did these studies in primary hepatocytes. They showed an increased palmitate incorporation into citrate.

1. The aspect concerning body temperature regulation is confusing. Would Itaconate not promote fatty acid oxidation to increase or maintain body temperature? Itaconate must therefore not be involved in the hypothermic response? Bringing UCP1 into the finding is confusing and needs to be better explained. Again a diagram would help, but enhanced BAT fatty acid oxidation and UCP1 expression appear linked here, with both being affected by Itaconate. This needs clarifying.

We appreciate this comment. The rationale is that if itaconate is stabilizing fatty acid oxidation, it would be necessary to fuel thermogenesis, a process dependent on fatty acid utilization. Our data support a role for itaconate in stabilizing body temperature following inflammation, potentially through enhanced fatty acid oxidation. This is evidenced by the hypothermic response to LPS in Acod1 KO mice. Furthermore, Mills et al. Nature 2018 show 4-OI injection boosts body temperature following LPS stimulation.

**Reviewer #2:**
Some conclusions involving the Irg1 knockout mice require important controls and clarifications to be fully convincing and some controls are missing.

We appreciate the needs for appropriate controls. Negative controls were omitted when baseline phenotypes were not observed. Due to time and resource limitations we were unable to repeat the experiments.